# Spatial Reorganization of Liquid Crystalline Domains of Red Blood Cells in Type 2 Diabetic Patients with Peripheral Artery Disease

**DOI:** 10.3390/ijms231911126

**Published:** 2022-09-22

**Authors:** Giada Bianchetti, Gaetano Emanuele Rizzo, Cassandra Serantoni, Alessio Abeltino, Alessandro Rizzi, Linda Tartaglione, Salvatore Caputo, Andrea Flex, Marco De Spirito, Dario Pitocco, Giuseppe Maulucci

**Affiliations:** 1Department of Neuroscience, Biophysics Section, Università Cattolica del Sacro Cuore, 00168 Rome, Italy; 2Fondazione Policlinico Universitario “A. Gemelli”, Istituto di Ricovero e Cura a Carattere Scientifico (IRCCS), 00168 Rome, Italy; 3Diabetes Care Unit, Università Cattolica del Sacro Cuore, 00168 Rome, Italy

**Keywords:** diabetes, red blood cells, macrovascular complications, peripheral artery disease, membrane fluidity, membrane microdomains, confocal microscopy, machine-learning, Laurdan, diabetic foot ulceration

## Abstract

In this work, we will investigate if red blood cell (RBC) membrane fluidity, influenced by several hyperglycemia-induced pathways, could provide a complementary index of HbA1c to monitor the development of type 2 diabetes mellitus (T2DM)-related macroangiopathic complications such as Peripheral Artery Disease (PAD). The contextual liquid crystalline (LC) domain spatial organization in the membrane was analysed to investigate the phase dynamics of the transition. Twenty-seven patients with long-duration T2DM were recruited and classified in DM, including 12 non-PAD patients, and DM + PAD, including 15 patients in any stage of PAD. Mean values of RBC generalized polarization (GP), representative of membrane fluidity, together with spatial organization of LC domains were compared between the two groups; *p*-values < 0.05 were considered statistically significant. Although comparable for anthropometric characteristics, duration of diabetes, and HbA1c, RBC membranes of PAD patients were found to be significantly more fluid (GP: 0.501 ± 0.026) than non-PAD patients (GP: 0.519 ± 0.007). These alterations were shown to be triggered by changes in both LC microdomain composition and distribution. We found a decrease in Feret diameter from 0.245 ± 0.281 μm in DM to 0.183 ± 0.124 μm in DM + PAD, and an increase in circularity. Altered RBC membrane fluidity is correlated to a spatial reconfiguration of LC domains, which, by possibly altering metabolic function, are associated with the development of T2DM-related macroangiopathic complications.

## 1. Introduction

Diabetes mellitus (DM) affects approximately 170 million people worldwide, with the global burden expected to rise to 366 million by 2030 [1,2]. Impaired insulin production or an insufficient response to released insulin are the main causes of DM [3]. Diabetes mellitus is a substantial risk factor for atherosclerotic occlusive disease of the lower extremities, also known as peripheral artery disease (PAD). PAD is linked to a higher risk of lower-extremity amputation and serves as a marker for atherothrombosis in the cardiovascular, cerebrovascular, and renovascular systems. As a result, patients with PAD have a higher risk of Myocardial Infarction (MI), stroke, and death [4]. In diabetic people, PAD also causes severe long-term disability [5]. PAD is an independent risk factor for ulceration and limb loss. It is present in 50% of patients with diabetic foot ulceration (DFU), a proportion that may increase [6,7]. Those with DFU and PAD are less likely to heal and more likely to require amputation compared to patients without PAD. Because of the necessity for a multitude of diagnostic testing, therapeutic procedures, and hospitalizations, treating individuals with PAD can be costly [8]. It is therefore essential that PAD is identified in all patients with diabetes. In individuals with pre-existing DM, age, duration of diabetes, smoking, and peripheral neuropathy are all linked to an elevated risk of PAD [9]. Hyperglycemia, dyslipidemia, and insulin resistance promote PAD formation and progression by processes similar to those found in coronary and carotid artery disease [10]. Glycosylated hemoglobin (HbA1c) is an essential predictor of long-term glycemic control and corresponds well with the risk of long-term diabetic complications such as PAD [11,12]. The Diabetes Control and Complications Trial (DCCT) found that rigorous therapy targeted at maintaining near-normal blood glucose levels significantly lowers the risk of PAD development or progression [13]. However, it is unclear why some patients with strong HbA1c levels develop PAD, whereas others with poor glycemic control remains unaffected even after years of illness. Other hyperglycemia-related physiopathological mechanisms and some independent risk factors, including hypertension and dyslipidemia, are believed to play a crucial role in the onset and progression of PAD, in addition to nonenzymatic glycosylation of proteins.

Recent research pointed out that these factors can indeed alter the physical state of biological membranes [14,15]. This is described in terms of phases, being homogenous regions of lipids with the same physical state and lipid composition. In the membrane, lipids can be laterally ordered and closely packed together in a liquid crystalline phase (LC) or in a less ordered fluid state, which is distinguished by a greater area filled by polar heads of phospholipids. Typically, a mosaic of gel and fluid domains made up of different phases coexist in the plane of the bilayer [16,17]. A quantifiable quantity called fluidity, whose value depends on the phase, curvature, and microviscosity of biological membranes as well as their phase, lipid structure, packing, and composition, describes the phase heterogeneity of biological membranes [18,19]. Because of post-translational alterations, red blood cell (RBC) plasma membrane fluidity could provide a supplementary index to HbA1c, representing the activation of metabolic pathways potentially associated with the development of microvascular problems. We have already shown that T1DM patients with complications have higher membrane fluidity with respect to non-complicated patients levels, while HbA1C levels were not statistically significant [20,21]. These observations suggest that RBC membrane fluidity is, with respect to HbA1c, an independent marker representative of risk factors contributing to the development of vascular complications. Therefore, pathological events characteristic of T1DM other than long-term hyperglycemia, particularly those related to diabetic dyslipidemia, may change the physical properties of the RBC membrane [22,23].

The goal of this study is to extend our results, previously reported for T1DM and microangiopathic complications, also to T2DM and macroangiopathic complications, and to investigate by means of confocal microscopy how the spatial organization of LC domains is altered. We investigated if membrane fluidity in T2DM patients can provide a more selective index for distinguishing patients with PAD (DM + PAD), allowing for the development of a new PAD marker in T2DM patients. We used laser functional confocal microscopy, a fluidity monitoring technique that can produce RBC fluidity maps on a sub-micrometric scale. The fluorescent probe Laurdan, which is equally distributed in the plasma membrane (PM), generates fluorescence whose colour is determined by the lipid packing of coexisting locations in the PM, allowing for quantitative contrast. Unlike other methods for retrieving PM fluidity changes, this method allows solid-ordered and liquid-disordered domains on PM to be clearly shown at their true length scale [19].

## 2. Results

### 2.1. Characteristics of the Population under Study

27 diabetic individuals, aged between 50 and 85 years (mean = 72.5 ± 7.9 years) were enrolled in this study, among which 23% were women, and 77% men (χ^2^ = 0.060). Their epidemiological features are summarized and compared with those of 10 healthy subjects (CTRL) in Table 1. The three groups were comparable for age (mean values = 65.8 ± 8.3 for CTRL, 72.0 ± 6.5 for DM and 72.9 ± 9.0 for DM+PAD, *p*-value = 0.09), BMI (*p*-value = 0.53), triglycerides (*p*-value = 0.98), creatinine (*p*-value = 0.16), and smoke habits (Yes = smoker, No = not smoker, χ^2^ = 0.91). CTRL subjects showed significantly lower levels of HbA1c (5.6% ± 0.3%) with respect to both DM (6.5% ± 0.7%, ** *p*-adj = 0.003) and DM+PAD (7.3% ± 1.6%, * *p*-adj = 0.01). CTRL subjects were characterized by a different lipid profile with respect to DM and DM+PAD. Levels of total cholesterol as well as HDL and LDL were significantly higher in CTRL subjects, who were not on any lipid-lowering therapy.

Among the diabetic populations, diabetes duration is comparable (*p*-value = 0.07) for DM (19.6 ± 6.0 years) and DM+PAD subjects (24.3 ± 7.0 years), as well as total cholesterol (*p*-value = 0.95), high-density lipoproteins (HDL, *p*-value = 0.50), low-density lipoproteins (LDL, *p*-value = 0.88), levels of creatinine (*p*-value = 0.13), and glycated hemoglobin (HbA1c), with values of 6.5% ± 0.7% for DM, and 7.3% ± 1.6% for DM+PAD, respectively (*p*-value = 0.15). 

### 2.2. Evaluation of RBC Plasma Membrane Fluidity

The sub-micrometric phase state maps of CTRL (Figure 1A), DM (Figure 1B) and DM+PAD (Figure 1C) RBC are shown in Figure 1, along with the distribution of GP values in the three groups (Figure 1D). In Figure 1A–C, each pixel is colored based on its GP value, on a scale ranging from white (gel-like phase, low fluidity, high GP = 0.60) to blue (liquid-like phase, high fluidity, low GP = 0.01) to blue (liquid-like phase, high fluidity, low GP = 0.01).

Because RBC lack inner organelles or nuclei, Laurdan labeled the plasma membrane specifically, thus providing extensive information regarding the arrangement of micro-domains, as can be observed in RBC from healthy subjects (Figure 1A) and from diabetic patients without PAD (Figure 1B) where the most LC domains are preserved, and in RBC from subjects with diagnosed PAD (Figure 1C), revealing the presence of yellow-colored regions, corresponding to more fluid domains. The quantitative analysis of changes in RBC membrane fluidity, was reported in the box plot in Figure 1D, showing the distribution of GP values of the three groups under analysis. GP, which provides an index of membrane fluidity associated to each patient, has a mean value of 0.534 ± 0.018 in CTRL (in green), being significantly higher with respect to both DM (in yellow, GP = 0.519 ± 0.007, * *p*-adj = 0.03) and DM+PAD (in purple, GP = 0.501 ± 0.026, ** *p*-adj = 0.008). The statistical comparison of these values, evaluated through the Anova test followed by a post-hoc comparison, revealed a significant difference in RBC membrane fluidity of diabetic subjects depending on the presence or not of diabetes-related PAD (*p*-adj = 0.04). Interestingly, DM and DM+PAD patients were comparable for plasmatic levels of HbA1c, a nonenzymatic glycosylation product correlated with an increased risk of developing DM-related microangiopathic complications, whose values were not correlated with GP (R^2^ = 0.29).

### 2.3. Characterisation of Sub-Micrometric LC Domains in RBC Membrane

To further evaluate and characterize differences in lipid domains structure in RBC from CTRL, DM, and DM+PAD subjects, we isolated LC compartments from the GP maps, according to the workflow explained in detail in Materials and Methods and showed in the following Figure 2.

In Figure 2A,C,E, the GP map of a CTRL, a DM, and a DM+PAD RBC, respectively, is represented in a color-coded scale, reported along with the image, which ranges from blue (low GP, high fluidity) to white (high GP, low fluidity). According to this scale, each pixel’s color in the GP map thus reflects the sub-micrometric fluidity level of the environment. A comparison between Figure 2A, Figure 2C, and Figure 2E allowed observing the presence of white regions in CTRL subjects, as well as a shift towards red and magenta colors in DM RBC, while a higher presence of yellow-green pixels characterized the map of DM+PAD subjects. To deeper evaluate this difference, which has been previously observed also in the mean distribution of GP values (Figure 1), we focused on the characterization of LC domains in healthy and T2DM patients with or without PAD by isolating them depending on the fluidity distribution retrieved in DM RBC. The described procedure allowed obtaining a binary mask for LC domains in CTRL (Figure 2B), DM (Figure 2D), and DM+PAD (Figure 2F) RBC, respectively, reported along with a magnification (dotted green frame). Differences, which can be qualitatively observed from both the GP and binary maps, were further quantified, analyzing the distribution of the mean number, the Feret diameter and the circularity of LC domains for CTRL, DM, and DM+PAD RBC, respectively, and the results are reported in the following graphs (Figure 3).

The box plot in Figure 3A reports the number of LC domains identified for each RBC (with n = 20) in CTRL (in green), DM (in yellow), and DM+PAD (in purple) subjects, respectively. As highlighted by the Tukey’s post-hoc comparison, the number of domains was comparable among the diabetics, with or without PAD (*p*-adj = 0.368), with mean values of 95.4 ± 10.4 for DM and 100.1 ± 14.9 for DM+PAD, respectively. Conversely, CTRL subjects (in green) are characterized by a significantly lower number of domains, with mean value of 20.2 ± 6.4, and *p*-adj < 0.0001 with respect to both DM and DM+PAD.

Interestingly, although a comparable number of domains among diabetic patients, significant differences were observed in the size and shape of micro-domains. To characterize detected domains, we evaluated the Feret diameter, representing the longest distance between any two points along the selection boundary, and the circularity, which equals 1 in case of perfect circle, and approaches 0, with increasingly elongated shape. Violin plots for Feret diameter and circularity are reported for each group in Figure 3B and 3C, respectively. Looking at these graphs, we can observe that CTRL RBC are characterized by the highest number of elongated LC domains, with mean circularity = 0.643 ± 0.368 and Feret diameter = 1.028 ± 1.655 μm. A progressive trend towards the presence of smaller and more circular LC domains can be then retrieved in DM and DM+PAD subjects, respectively, which are characterized by values of circularity closer to 0 (0.846 ± 0.242, for DM, and 0.903 ± 0.182 for DM+PAD, Figure 3C, *p*-adj < 0.0001) and a longer Feret diameter (0.245 ± 0.281 μm for DM and 0.183 ± 0.124 μm for DM+PAD, Figure 3B, *p*-adj < 0.0001).

## 3. Discussion

In this study, we looked at the changes in the Laurdan emission spectrum in a sample of T2DM subjects with a long history of the disease (>10 years) by measuring GP, representative for RBC membrane fluidity. A statistically significant difference was discovered between the T2DM and T2DM with PAD. The membranes of PAD patients’ RBC were found to be more fluid than those of non-PAD patients. HbA1c values were not statistically significant between the two groups (*p*-value = 0.15), despite being a frequently used marker to assess the onset of diabetes-related comorbidities. This finding implies that the HbA1c test may not account for all risks that contribute to the development of cardiovascular complications, which instead include modifications on the physical state of the RBC membrane. As a result, this assay may be a useful complement of HbA1c in determining the risk to develop diabetes-related complications, including PAD and DFU.

We previously observed increasing fluidity, a phase separation, and a formation of micro-domains with different fluidity in RBC membranes of T1DM patients and T1DM patients with complications [18,19,20,21]. We showed that the GP index is able to not only monitor the progression of T1DM, but also discriminate the development of complications [22]. Here, we showed that GP is not only a marker of DR in T1DM patients but also a marker of PAD development in T2DM patients. These alterations are triggered, from a microscopic point of view, by alterations in both microdomain composition and distribution. In this context, by exploiting information embedded in the images, we analysed in detail how the induced phase separation observed in the PAD group consists of the shrinking of the liquid crystallin domains rather than in the reduction in their number. Indeed, the number of LC cluster per cell is not significantly different between the two groups, whereas the mean value of Feret diameter decreases from 0.245 ± 0.281 μm to 0.183 ± 0.124 μm and the circularity increases from 0.846 ± 0.242 to 0.903 ± 0.182, indicating an abrupt shape change. The less extension of LC micro-domains that are preserved in patients without microvascular complications could be not only an epiphenomenon, but it could also be related to the physiopathology of early stages of diabetic microangiopathy (capillary occlusion, micro-aneurysm formation, retinal ischemia). Membrane fluidity can integrate in a more stable fashion modifications induced by dyslipidemia, as assessed by standard measures, including raised plasma triglycerides and LDL-cholesterol, in the context of decreased HDL-cholesterol. It is well known that these lipids alone cannot accurately report the complex dyslipidemia associated with both types of diabetes. Dyslipidemia is frequently related to T2DM, with increased triglycerides and low HDL cholesterol as the main manifestations. Abnormalities in lipid management are rarely linked to T1DM, except for an increase in HDL cholesterol in some patients. Dyslipidemia usually appears later in T1DM, along with the onset of renal impairment. However, one major study has shown lipid abnormalities in T1DM patients in the early stages [24]. Individuals with T1DM showed lower serum levels of succinic acid and phosphatidylcholine (PC) at birth in that study. Hyperlipidemia, in general, is associated with increased uptake of free fatty acids by cells. CD36 and members of the fatty acid binding protein (FABP) family are the most common mediators of fatty acid absorption into tissues [25]. Furthermore, in diabetics, circulating soluble CD36 quantities are greater [26]. Serum A-FABP and E-FABP concentrations have also been recognized as possible biomarkers for nephropathy and related cardiovascular risk in people with T2DM [27]. We can hypothesize that alterations of these absorption mechanisms, and the consequent alteration of RBC lipidome, along with systemic effects related to the chronic hyperglycemia, oxidative stress, and metabolic alterations triggered by the systemic state (glyco-oxidation), can lead to the observed alterations of membrane fluidity and LC domain distribution in both forms of diabetes mellitus. Moreover, changes in the composition or distribution of lipids in the membranes of erythrocytes could result in abnormalities in the membrane deformability, leading to a reduced cell life span or to an altered blood rheological pattern, which can play a role in the PAD pathogenesis [28]. Further experiments and investigations are needed to assess which pathways or mechanisms are influenced by this transition.

However, the use of RBC membrane as a more stable index for dyslipidemia is reliable: in literature, several evidences are reported of impaired lipid composition of plasma membranes in insulin-sensitive tissues such as liver, muscle, and adipose tissue of patients affected by metabolic syndrome [29] and diabetes. This impairment can be reflected in RBC membrane fluidity, functioning as a systemic biosensor of other tissues’ state. It is worthwhile to point out that beta cells, vascular cells, and RBC could be altered in different directions, being characterised by a different lipid homeostasis. Indeed, mature RBC lack nuclei and other cell organelles, as well as the ability to generate and change proteins in post-translational processes. This aspect, on the other hand, confers to RBC a unique sensitivity to variations in lipid composition, oxidative stress, and glucotoxicity-induced post-translational modifications, which can modulate the distribution of LC domains [30,31,32,33,34,35].

In literature, conflicting data about RBC membrane fluidity report either reduced, unchanged or increased fluidity in diabetes [36]. In these reports, the most typical measurements of fluidity are performed using the hydrophobic probe DPH (1,6-diphenyl-hexa-1,3,5-triene). However, while DPH measures the apparent viscosity of the side chain region (centre) of membranes, Laurdan compartmentalizes at the hydrophilic-hydrophobic membrane interface [37] thus monitoring the phase of a different region of PM. By combining spin labelling and electron spin resonance measurements, Kamada and Otsuji in 1983 came at a similar conclusion [38]. Using probes spanning different regions of the membranes, they found that the decrease in fluidity was located only in deeper sites (hydrophobic region) of the erythrocyte membrane in diabetic patients. We found a similar behavior in PC-12 cells with Laurdan derivatives engineered with increased chain length [39]. Overall, this method has, as an additional advantage, the possibility to detect the presence of complications after a blood draw without requiring a time-consuming or expensive analysis process or a specialist visit. Furthermore, because confocal microscopes are widely available in hospital research facilities, the technique can be adapted into clinical settings in a reasonably quick timescale after its definitive validation. The study’s main disadvantage is its small sample size, which necessitates further testing on a larger population. Moreover, there is a danger of bias caused by other environmental factors that may influence membrane fluidity. However, this is not the first study in which we found this alteration in diabetic subjects [15,21]. To overcome both these limits, the capacity to obtain large-scale fluidity maps could enable the creation of algorithms and the selection of distinct fluidity patterns depending on a variety of clinical parameters.

## 4. Materials and Methods

### 4.1. Patients’ Recruitment

Ten healthy subjects and 27 patients with T2DM were enrolled in the Diabetes Care Unit at Fondazione Policlinico Universitario “A. Gemelli”, IRCSS (Rome, Italy). Subjects with T2DM were divided into two groups: those without PAD (DM, n = 12) and those with PAD at any stage of the disease (DM+PAD, n = 15). T2DM diagnosis, long duration of disease (>10 years), age between 50 and 85 years, and BMI < 35 kg/m^2^ were the inclusion criteria for the CTRL, DM, and DM+PAD groups. T1DM, steroid-or drug-induced diabetes, diabetes associated with other endocrinopathies, maturity-onset diabetes of the young (MODY), chronic pancreatitis or previous pancreatectomy, a history of cancer in the previous 5 years, hematologic disorders associated with hemolysis, and gestational diabetes were all exclusion criteria for the two groups. The ethical committee approved this study, and all clinical trials followed the Declaration of Helsinki’s principles. All patients gave their informed consent for study participation.

### 4.2. Sample Preparation and Measurements

RBC were extracted as previously reported [20], seeded on a multi-well plate, and directly observed under the microscope. RBC were stained with the fluorescent probe Laurdan (6-Dodecanoyl-2-Dimethylaminonaphthalene), which is characterised by a shift in the emission spectrum reflecting the lipid phase state of the environment (bluish in ordered, gel phases and greenish in disordered, liquid-crystalline phases) to assess membrane fluidity [40]. A Nikon A1-MP confocal microscope was used to image RBC within 1 h from extraction. Laurdan (Molecular Probes, Inc, Eugene, OR, USA) was prepared as a 1 mmol/L stock solution in DMSO and then added to RBC in 0.9 percent NaCl (dilution 1:1000). Before imaging, stained RBC were incubated for 30 min at 37 °C with 5% CO_2_.

For each patient, five images (about 100 RBC) of Laurdan emission intensity were acquired concurrently in two distinct channels (emission filter: 450/50 nm for the blue channel, 525/50 nm for the green channel) using a 60× immersion-oil objective with 1.4 NA for the estimation of generalized polarization (GP). The imaging was done at 37 °C with 5% CO_2_ (cage incubator, OKOLAB).

To separate RBC from the backdrop and eliminate debris and other aggregates, we used the open-source software Ilastik [41] to do a fine-tuned, supervised pixel classification. The RBC segmentation workflow is described in depth in [15]. RBC masks were then applied on the original dual-channel pictures, and the GP index was determined for each pixel using the equation below:(1)$$GP=\frac{I_{blue}-GI_{green}}{I_{blue}+GI_{green}}$$
where $I_{blue}$ and $I_{green}$ are the intensities for the blue and green channel, respectively, and G is a calibration factor that depends on the experimental setup.

### 4.3. Isolation and Analysis of RBC Membrane LC Domains

RBC membrane domains were isolated through the open-source software ImageJ (NIH), according to the following workflow:

A stack consisting of 20 GP maps of RBC from DM subjects was created and the stacked GP distribution, shown in Figure 4, was extracted. Values of GP belonging to the highest quartile (75 percentile, GP > 0.582) were considered representative of LC domains.

Using the GP reference value, a threshold was set for each RBC from both DM and DM+PAD subjects, and a binary mask representing LC domains was thus obtained, as shown in Figure 5.

Through the *Analyze Particle* tool (ImageJ, NIH), we quantified the number of domains for single RBC, the Feret diameter, which can be defined as the longest distance between any two points along the selection boundary (also known as maximum caliper), and the circularity, calculated as:(2)$$Circ=\frac{4\pi \cdot A}{p^{2}}$$
where *A* and *p* indicate the area and the perimeter of the domain, respectively. A circularity value of 1 is representative of a perfect circle, whereas as the value approaches 0, it indicates an increasingly elongated shape.

### 4.4. Statistics

Statistical tests on sets of biological/biophysical data were performed using Python open-source *SciPy* library (https://scipy.org/, accessed on 21 September 2022). The normality of the data was visually verified using quartile-quartile plots. Anova test, followed by a post-hoc comparison, was used to compare data among the three groups. *p*-values < 0.05 were regarded as statistically significant.

## 5. Conclusions

Membrane fluidity, as quantified by the GP index retrieved by confocal metabolic imaging techniques [14,15], is an optimal candidate in complementing HbA1c in both long-term T1DM and T2DM management, allowing a more sensitive assessment of the progression of dyslipidemia. This investigation extended our results obtained on T1DM and microangiopathic complications to T2DM and macroangiopathic complications, and constitutes the first evidence that the molecular mechanisms leading to microvascular and macrovascular complications can be similar in these two forms of diabetes. These mechanisms include changes in lipid composition of RBC along with a network of systemic effects related to chronic hyperglycemia, oxidative stress, and metabolic alterations, which have as an effect a LC domain reorganization that can have a functional role, which remains to be deeply investigated. Large scale collection of fluidity maps will allow both to clarify this point, and to select distinct fluidity patterns, thus increasing the specificity of this assay in identifying the factors leading to fluidity variation in different clinical contexts.

## Figures and Tables

**Figure 1 ijms-23-11126-f001:**
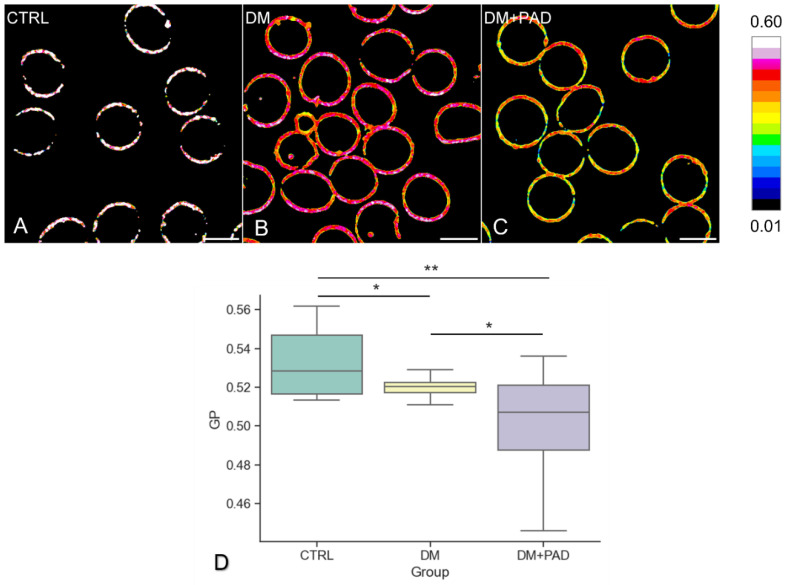
**Representative fluidity maps and distribution of GP values for CTRL, DM, and DM+PAD RBC.** In (**A**–**C**), each pixel’s color represents the Generalized Polarization (GP) values, which vary from white (low fluidity, high GP = 0.60) to blue (high fluidity, low GP = 0.01), according to the LUT bar reported along with the images. The preponderance of most LC (white and pink) domains is distinguishable in CTRL (**A**), and DM (**B**) RBC, whereas DM+PAD (**C**) patients are characterized by more fluid domains (colored in yellow and green). 5 images (about 100 RBC) were evaluated for each patient. Scale bar is 5 μm. The box plot in (**D**) shows the distribution of generalized polarization values calculated for each of the 37 subjects under study: CTRL (in green, on the left), DM (in yellow, in the middle) and DM+PAD (in purple, on the right). The three groups were statistically different for mean values of GP, according to Anova test followed by a post-hoc comparison (*p*-adj = 0.03 for CTRL-DM; *p*-adj = 0.008 for CTRL-DM+PAD; *p*-adj = 0.04 for DM-DM+PAD). Holm method was used to adjust the *p*-value (** stands for *p* < 0.01; * stands for *p* < 0.05).

**Figure 2 ijms-23-11126-f002:**
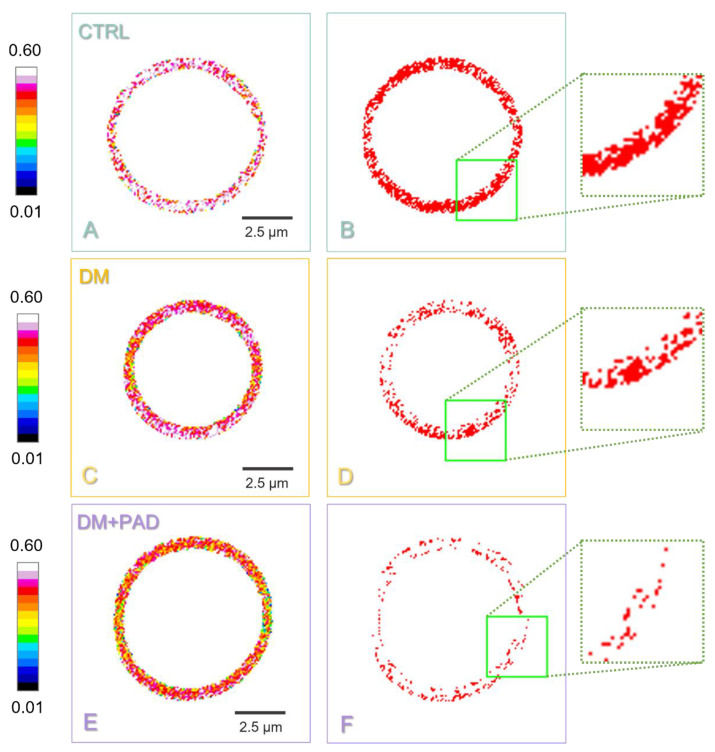
**Identification of sub-micrometric LC domains in CTRL, DM, and DM+PAD red blood cells.** The GP map of CTRL, DM, and DM+PAD RBC is reported in (**A**,**C**,**E**), respectively. Each pixel’s color reflects its GP value, according to the LUT reported along with the image (from blue for low GP—high fluidity, to white for high GP—low fluidity). Scale bar is 2.5 μm. A binary mask representing LC domains is shown in (**B**,**D**,**F**) for CTRL, DM, and DM+PAD RBC, respectively. From this analysis, a fine characterization of the number, size and shape of LC domains can be obtained for each RBC. A magnification (dotted green frame) of domains is represented to allow qualitatively appreciating differences between CTRL, DM, and DM+PAD subjects, where a progressive trend towards the presence of smaller LC domains can be observed.

**Figure 3 ijms-23-11126-f003:**
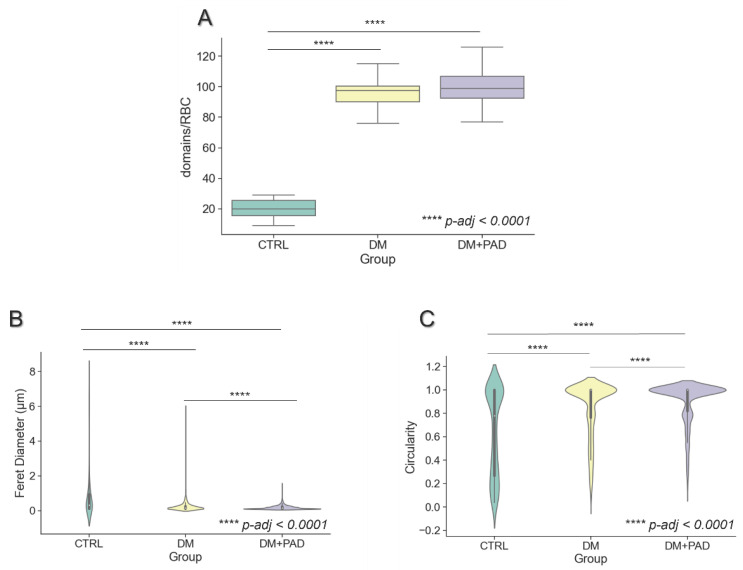
**Quantitative analysis of LC domains in CTRL, DM, and DM+PAD RBC.** (**A**) The box plot shows the distribution of number of LC domains calculated for 20 RBC of CTRL (on the leftm in green), DM (in the middle, in yellow), and DM+PAD (on the right, in purple). Mean values are comparable among diabetic subjects (95.4 ± 10.4 for DM and 100.1 ± 14.9 for DM+PAD, respectively, *p*-adj = 0.368), and they result significantly higher with respect to healthy controls (20.2 ± 6.4, *p*-adj < 0.0001). (**B**) The violin plot represents the distribution of LC domains Feret diameter (in μm), evaluated for CTRL (in green), DM (in yellow), and DM+PAD (in purple) RBC. From the graph, a significant difference in LC domains size (*p*-adj < 0.0001) among the three groups can be observed. In (**C**) the violin plot for domains’ circularity is reported for CTRL (in green), DM (in yellow) and DM+PAD (in purple), respectively. This analysis shows that LC domains in DM+PAD RBC are characterized by a higher circularity (*p*-value < 0.0001), while a trend towards a more elongated shape can be retrieved in DM and CTRL subjects, respectively.

**Figure 4 ijms-23-11126-f004:**
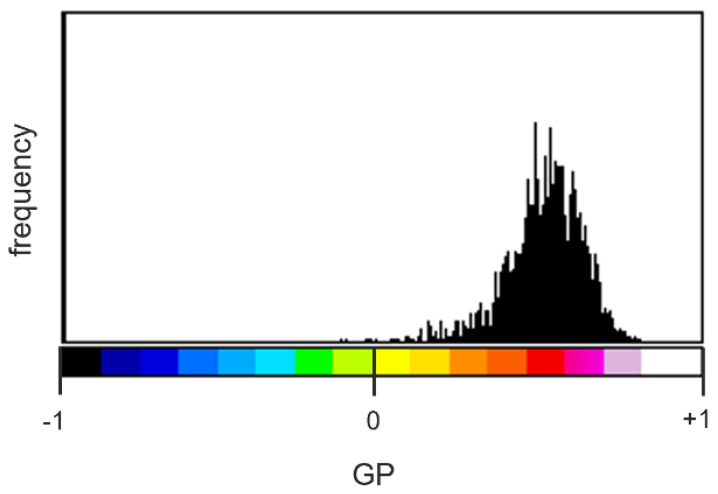
**Stacked GP distribution of DM RBC.** The graph shows the number of pixels (frequency, on the *y*-axis) as a function of GP (on the *x*-axis) evaluated for a stack of 20 RBC from DM subjects. The LUT reported along with the histogram allowed associating a colour to each pixel, representing the corresponding value of fluidity.

**Figure 5 ijms-23-11126-f005:**
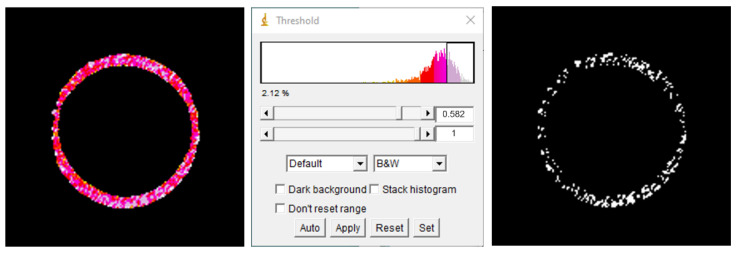
**Construction of the binary mask representing LC domains in RBC membrane.** Setting a proper threshold, corresponding in our case to values of GP belonging to the highest quartile (histogram, in the centre), allowed isolating domains from the GP map (represented in a color-coded scale on the left) and obtaining a binary mask of LC domains for each RBC (on the right).

**Table 1 ijms-23-11126-t001:** Baseline characteristics of the 10 healthy and 27 diabetic subjects under study.

Variable	CTRL, N = 10 ^1^	DM, N = 12 ^1^	DM+PAD, N = 15 ^1^	*p*-Value ^2^	Post-hoc Comparison (*p*-adj ^3^)		
					CTRL-DM	CTRL-PAD	DM-PAD
Duration of diabetes (y)	-	19.6 ± 6.0	24.3 ± 7.0	<0.0001	<0.0001 (****)	<0.0001 (****)	0.08
Age (y)	65.8 ± 8.3	72.0 ± 6.5	72.9 ± 9.0	0.09			
HbA1c (%)	5.6 ± 0.3	6.5 ± 0.7	7.3 ± 1.6	0.006	0.003 (**)	0.01 (*)	0.15
BMI (kg/m^2^)	26.5 ± 3.2	24.0 ± 4.8	26.2 ± 8.0	0.53			
Total Cholesterol (mg/dL)	203.0 ± 37.6	138.3 ± 52.8	137.1 ± 43.6	0.002	0.009 (**)	0.002 (**)	0.95
HDL (mg/dL)	64.7 ± 21.2	37.5 ± 15.6	41.4 ± 12.4	0.001	0.008 (**)	0.008 (**)	0.50
LDL (mg/dL)	111.5 ± 23.0	72.7 ± 43.3	70.1 ± 39.9	0.01	0.03 (*)	0.01 (*)	0.88
Triglycerides (mg/dL)	118.6 ± 83.2	125.3 ± 107.4	123.8 ± 36.5	0.98			
Creatinine (mg/dL)	0.73 ± 0.12	0.80 ± 0.35	1.92 ± 2.36	0.16			
Smoke				0.91			
Yes	0/10 (0%)	2/12 (17%)	1/15 (7%)				
No	10/10 (100%)	10/12 (83%)	14/15 (93%)				

^1^ Mean ± SD or Frequency (%). ^2^ Anova or χ^2^. ^3^
*p*-adjusted (Holm method). **** stands for *p* < 0.0001; ** stands for *p* < 0.01; * stands for *p* < 0.05.

## Data Availability

Not applicable.

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
