# Peer review of "Spatial Reorganization of Liquid Crystalline Domains of Red Blood Cells in Type 2 Diabetic Patients with Peripheral Artery Disease"

_ijms, 2022, doi:10.3390/ijms231911126_

Round 1

Reviewer 1 Report

Bianchetti et al. manuscript is interesting and explores the idea of measuring membrane order being a more way to evaluate T2D patients. The manuscript has potential. However, it includes few experiments, lacks proper controls and a second technique to support the conclusions.

- The manuscript lacks proper controls: i.e. assaying in parallel  RBC from healthy individuals. This is an important concern and must be addressed experimentally.

- The manuscript will benefit of a more direct and concise tittle.

- Was it expected to find RBC from patients with PAD to be more fluid? How does it fit the literature (non-limited to own lab results)?

- N number needs to be clarified in the figure legends. I.e. in Fig. 2 n=12-15 patients. But, how many images were quantified for each patient?

- Color scales must show numbers (not only high/low fluidity - order will be better). I.e. in figure 1 and 3. It is hard to judge without numbers in the scale, but images in figure 1 do not appear to represent the media/mean obtained in figure 2, where the differences in GP were about 0.01-0.02. In addition, authors should try to maintain similar ranges across figures i.e. figures 1, 3 and 5.

- Figure 5 shows that all the pixels were towards +1 (scale -1 to +1). Therefore limiting the ability to detect deviations towards more rigid. A difference microscope configuration would have been preferred. 

- Figure 2 could be panel B and C of figure 1.

- Statistics needs to be clarified in Table 1. Why ANOVA was used and not always t-test? There are only two groups DM and PM+PAD. Which type of ANOVA? What parameters were analysed by ANOVA?

- Citations/references must be revised. At least 14/45 references include authors of the manuscript under evaluation. Please, limit to the most relevant ones. Authors could also consider to discuss the work of Prof. Perona (i.e. doi: 10.1016/j.bbamem.2017.04.015) and Prof. Pilon (doi: 10.1186/s12944-016-0342-0).

Reviewer 2 Report

see file

Round 2

Reviewer 1 Report

Thanks for the updated manuscript.

The manuscript still lacks proper controls: i.e. assaying in parallel  RBC from healthy individuals in all the experiments. This is an important concern and must be addressed experimentally. Now controls are added in Figure1 and only shown in a supplementary figure. What about the other figures/experiments? Were control samples included? Were the controls and DM patients analysed simultaneously?

Authors mention that the work from Prof. Pilon is discussed. However, his work is not cited in the main text. Several citations can be found in his review about RBC from DM patients (typically T2D) have increased membrane rigidity. All those antecedents are worth to discuss, particularly when previous literature does not match the results in the manuscript.

Please revised smoking % in table S1. Table S1 should be Table 1. This reviewer does not understand why the controls are only shown in supplementary. Controls are fundamental in science and should be included in all the experiments.

Reviewer 2 Report

The authors have conveniently replied to all my comments.

 I have found comments from the other reviewer also pertinent and I judge that these were considered also.
